# Obtaining 2,3-Dihydrobenzofuran and 3-Epilupeol from *Ageratina pichinchensis* (Kunth) R.King & Ho.Rob. Cell Cultures Grown in Shake Flasks under Photoperiod and Darkness, and Its Scale-Up to an Airlift Bioreactor for Enhanced Production

**DOI:** 10.3390/molecules28020578

**Published:** 2023-01-06

**Authors:** Mariana Sánchez-Ramos, Silvia Marquina-Bahena, Laura Alvarez, Antonio Bernabé-Antonio, Emmanuel Cabañas-García, Angélica Román-Guerrero, Francisco Cruz-Sosa

**Affiliations:** 1Department of Biotechnology, Metropolitan Autonomous University-Iztapalapa Campus, Av. Ferrocarril de San Rafael Atlixco 186, Col. Leyes de Reforma 1a. Sección, Alcaldía Iztapalapa, Mexico City 09310, Distrito Federal, Mexico; 2Chemical Research Center-IICBA, Autonomous University of the State of Morelos, Av. Universidad 1001, Col. Chamilpa, Cuernavaca 62209, Morelos, Mexico; 3Department of Wood, Pulp and Paper, University Center of Exact Sciences and Engineering, University of Guadalajara, Km 15.5 Guadalajara-Nogales, Col. Las Agujas, Zapopan 45100, Jalisco, Mexico; 4Scientific and Technological Studies Center No. 18, National Polytechnic Institute, Blvd. del Bote 202 Cerro del Gato, Ejido La Escondida, Col. Ciudad Administrativa, Zacatecas 98160, Zacatecas, Mexico

**Keywords:** medicinal plant, anti-inflammatory compounds, cell cultures, airlift bioreactor, photoperiod, absolute darkness

## Abstract

*Ageratina pichinchensis* (Kunth) R.King & Ho.Rob. is a plant used in traditional Mexican medicine, and some biotechnological studies have shown that its calluses and cell suspension cultures can produce important anti-inflammatory compounds. In this study, we established a cell culture of *A. pichinchensis* in a 2 L airlift bioreactor and evaluated the production of the anti-inflammatory compounds 2,3-dihydrobenzofuran (**1**) and 3-epilupeol (**2**). The maximum biomass production (11.90 ± 2.48 g/L) was reached at 11 days of culture and cell viability was between 80% and 90%. Among kinetic parameters, the specific growth rate (*µ*) was 0.2216 days^−1^ and doubling time (*td*) was 3.13 days. Gas chromatography coupled with mass spectrometry (GC-MS) analysis of extracts showed the maximum production of compound **1** (903.02 ± 41.06 µg/g extract) and compound **2** (561.63 ± 10.63 µg/g extract) at 7 and 14 days, respectively. This study stands out for the significant production of 2,3-dihydrobenzofuran and 3-epilupeol and by the significant reduction in production time compared to callus and cell suspension cultures, previously reported. To date, these compounds have not been found in the wild plant, i.e., its production has only been reported in cell cultures of *A. pichinchensis*. Therefore, plant cell cultured in an airlift reactor can be an alternative for the improved production of these anti-inflammatory compounds.

## 1. Introduction

The secondary metabolites of plants play an important role in their development and survival, since they participate in the defense against insects, microorganisms, and other abiotic conditions [1,2,3,4,5,6]. However, the production of these metabolites is variable over time due to seasonal and environmental factors [7,8,9]. On the other hand, plants have been an important resource for humans due to their nutritional and medicinal attributes, such as antimicrobial, antioxidant, anti-inflammatory, antitumor, antiviral, antiparasitic, and antidiabetic [10,11,12,13,14,15,16,17,18]. Therefore, the interest in the use and study of medicinal plants and their bioactive metabolites has increased in the last decades since it also represents a viable field for the exploration of new bioactive molecules despite low yields [19,20].

In this regard, through the application of plant cell culture technology, it has been possible to obtain and improve the biomass and bioactive production of compounds, and there are several reports in which important results have been obtained, highlighting some advantages such as constant and controlled production in reduced spaces [21,22,23,24,25,26,27,28,29,30,31]. Moreover, using biotic and abiotic elicitors, it is possible to increase compound production; in particular, light conditions have been studied as an elicitor to increase biomass production, and bioactive compounds such as alkaloids and phenolics [32,33,34]. Another advantage of cell cultures is the ability to scale up to bioreactors to reduce culture times and increase the production of the compounds of interest. For instance, the production of taxol, resveratrol, berberine, diosgenin, vinblastine, and vincristine was significantly increased when cells were scaled up from flasks to bioreactors [35,36,37,38,39,40,41]; in fact, these are produced industrially [42,43,44,45]. These success stories show that it is possible to produce and even increase the production of compounds of interest in reactors, which contributes to the supply of necessary drugs.

Among other plants, *Ageratina pichinchensis* (Kunth) R.King & Ho.Rob. (Asteraceae) is an endemic species from the state of Morelos, Mexico, and it is used in traditional medicine as an antifungal, anti-inflammatory, and healing agent [46,47]. Previous studies have reported the production of triterpene, benzochromene, benzofuran, flavonoid, sterol, and eudesmane-type compounds, and their ethnomedical uses have already been validated [48,49,50,51,52,53,54,55].

In this regard, we have previously reported the establishment of callus cultures and cell suspension cultures of *A. pichinchensis* that maintained the ability to produce bioactive compounds. For instance, the (*2S*,*3R*)-5-acetyl-7,3α-dihydroxy-2β-(1-isopropenyl)-2,3-dihydrobenzofuran (**1**) compound (Figure 1A) has shown potential to inhibit the activation of NF-kB, IL-6, and TNF-α secretion as well as suppression of NF-kB factor expression that have been linked to disorders such as rheumatoid arthritis, chronic hepatitis, and pulmonary fibrosis [56,57]. On the other hand, the 3-epilupeol (**2**) compound (Figure 1B) has been attributed anti-inflammatory, anti-HIV, and anti-tuberculous effects [58,59,60]. For *A. pichinchensis*, the production of these compounds has only been obtained in callus cultures or cell suspension cultures [61,62].

This study aimed to cultivate cells of *A. pichinchensis* in shake flasks to evaluate the production of anti-inflammatory compounds under photoperiod and dark conditions, and then to use the best-established conditions to scale up the cell culture to an airlift bioreactor to increase production of the anti-inflammatory compounds **1** and **2**.

## 2. Results and Discussion

### 2.1. Cell Cultures in Shake Flasks under Photoperiod and Absolute Darkness

#### 2.1.1. Yield of Biomass

Biomass growth showed a similar behavior in its growth phases both under photoperiod and absolute darkness conditions (Figure 2A). For the cells cultured under photoperiod conditions, the adaptation phase was observed from 0 to 4 days, the exponential phase from 6 to 16 days, the stationary phase from 16 to 18 days in which the maximum dry-weight biomass (DW) was observed at day 18 (13.52 ± 0.44 g/L DW), and finally, the death phase was observed. The specific growth rate was 0.1281 days^−1^ with a doubling time of 5.41 days. For the cells cultured under absolute darkness conditions, the adaptation phase was on days 0 to 4, the exponential phase stopped at day 14 and then the stationary phase was maintained until day 20 and later the death phase was presented. Under these conditions, the maximum accumulation of biomass (10.33 ± 0.78 g/L DW) was also observed on day 18. In fact, the *µ* was 0.1518 days^−1^ and *td* was 4.57 days.

These results are similar to those previously reported for cultures of *A. pichinchensis* [62]. For instance, cell suspension cultures of *Ficus deltoidea* yielded high biomass production under photoperiod compared to dark condition [63]. Similarly, *Basella rubra* callus cultures exhibited higher biomass production when grown under photoperiod [64].

#### 2.1.2. Sugar Consumption, Cell Viability, and pH

Regarding sucrose levels, the behavior was very similar in both incubation conditions, observing a constant decrease from day 2 to 16, after which there was a total consumption until day 22 (Figure 2B). In addition, a relationship between biomass growth and sucrose consumption as a carbon source was observed.

On the other hand, cell viability had slight changes over culture time, when compared between both incubation conditions (Figure 2C). The initial cell viability (day 0) had an average 92% and decreased to 75.5% at day 22. This behavior is common in cultures reported in other species [64,65,66]. The pH of the culture medium is another relevant factor; some authors mention that pH value is a determining factor for secondary metabolite production [67,68,69]. In this case, the pH values had slight variation and remained between 5.6 and 5.2 for both incubation conditions (Figure 2D).

#### 2.1.3. Production of 2,3-Dihydrobenzofuran and 3-Epilupeol

The photoperiod stimulated the production of 2,3-dihydrobenzofuran and 3-epilupeol compounds (Figure 3), particularly during the exponential phase of culture growth. Higher production of 2,3-dihydrobenzofuran was achieved with photoperiod at day 8, obtaining 495.04 ± 22.85 µg/g DW (Figure 3A). Despite the same behavior occurring under dark conditions, the maximum production at day 8 was reduced by 36.3%.

Regarding 3-epilupeol, its production was closely related to crop growth, although no significant differences were observed by effect of photoperiod and absolute darkness (Figure 3B). The maximum production in photoperiod and absolute darkness was obtained during the stationary phase for both photoperiod and darkness treatments, with yields of 414.24 ± 31.56 and 395.14 ± 13.32 µg/g DW, respectively.

Cell suspension cultures have demonstrated to be suitable for improving the cell growth, producing a higher yield of biomass and not only higher contents but also great variety of bioactive molecules [70,71,72,73,74], when cell cultures are submitted to photoperiod conditions [62,75,76,77,78,79,80,81,82,83]. Similar results have been reported in cell suspension cultures for the production of antioxidant compounds in *Melastoma malabathricum* [84], lignans in *Linum usitatissimum* [85], and alkaloids in *Hyoscyamus muticus* [86], where higher contents were achieved when cultured under photoperiod conditions. Likewise, cell suspension cultures of *Rosa hybrida* cv. Charleston significantly enhanced the anthocyanin production when exposed to light conditions, but anthocyanin yield was drastically reduced when dark conditions were used [87]. *Tecoma stans* callus cultures also exhibited an increase in the production of antioxidant compounds when photoperiod conditions were used [88]. In this study, our results for *A. pichinchensis* showed that the biomass production and anti-inflammatory compounds in shaken flasks is favored under photoperiod conditions, whereby we decided to culture cells in an airlift bioreactor under the same conditions. However, some reports state that cell culture under darkness can promote the production of biomass and secondary metabolites, and even improve the morphology of cell suspensions by reducing cell aggregation and oxidation; however, this response depends on the type of compound looked for and the plant genotype [89,90,91,92,93,94,95].

### 2.2. Cell Culture in an Airlift Bioreactor under Photoperiod Conditions

#### 2.2.1. Growth Kinetics and Biomass Yield

*A. pichinchensis* cell culture was grown in the airlift bioreactor for 15 days, the adaptation phase was not observed, instead, the exponential growth was shown from day 0 to day 11, stationary phase lasted until day 12, and by day 13, the death phase was observed and lasted until day 15 (Figure 4A). The maximum biomass production occurred between days 11 and 12, with a yield of 11.90 ± 0.19 and 11.88 ± 0.44 g/L DW, respectively, time significantly shorter than that achieved for the cell culture in shake flasks. The growth index was 0.2216 days^−1^ and the doubling time lasted 3.13 days.

On the other hand, an inverse relationship between the consumption of the carbon source and biomass production was observed, where the carbon source was depleted at the end of the exponential phase, which agrees with the decrease in cell growth or the death phase; moreover, cell viability was maintained between 91.83 to 76.8% throughout the culture growth (Figure 4B).

Cell culture of *Haematococcus pluvialis* exhibited a specific growth rate of 0.31 days^−1^ [95]; meanwhile, cell cultures of *Taxus wallichiana* grown in a 20-L airlift bioreactor for 28 days reported a growth rate of 0.16 days^−1^ with cell viability between 100% and 85% [96,97]. *Solanum chrysotrichum* cells cultured in an airlift bioreactor for 21 days displayed a specific growth rate at 0.03–0.08 days^−1^, the pH remained between 5.7 and 5.3, and the cell viability was maintained between 100% and 70% [98]. Similar results were shown in *Panax notoginseng* cell cultures, in which, an enhancement in cell density and specific growth rate was obtained when cultured in an airlift bioreactor and modifying the composition culture medium conditions [99].

Regarding biomass production, no important changes were observed during the first three days of growth (Figure 5A), but after 4 days there was an increase in cell density and a slightly yellow coloration (Figure 5B). As growth increased, morphological changes in the cells were observed. Then, during the exponential phase the biomass was beige (Figure 5C).

Microscopic analysis with Evan’s blue showed circular aggregates made up of 5 to 9 cells (Figure 5D). Subsequently, the size of the aggregates increased in the stationary phase, observing more elongated cells and the viability also decreased (Figure 5E).

During crop growth, the pH was maintained at 5.5–5.8, avoiding sudden changes that could affect the cell culture. The formation of foam in the airlift bioreactor was controlled with the addition of antifoam agents [96] to reduce the limitations on the mass transfer processes.

#### 2.2.2. Production of 2,3-Dihydrobenzofuran and 3-Epilupeol

The extracts from the biomass cultivated in the airlift bioreactor showed that the production of 2,3-dihidrobenzofuran occurred from day 1, reaching its maximum production on day 7 with 903.02 ± 41.06 µg/g DW and gradually reducing its content until the end of the culture (Figure 6). This production is almost double that produced in flasks with only 495.04 ± 22.85 µg/g DW on day 8.

For the compound identified as 3-epilupeol, it was quantified from day 3, showing association with growth behavior, displaying a maximum production at day 14 with 561.63 ± 10.16 µg/g DW. Major production was observed in the exponential and stationary phase, which may be due to a response of the cells to nutrient limitations. The yielding of 3-epilupeol was significantly higher than that found in flask cultures (414.24 ± 31.56 µg/g DW). In a previous study of suspension cultures of *A. pichinchensis*, similar results were reported for these compounds, except for callus cultures where lower production was obtained [61,62]. Appendix A shows the chromatograms of the extracts of days 7 and 14, culture times where higher production of both compounds was found.

Table 1 shows a comparison of the production of compounds **1** and **2** obtained in different cell culture production systems of *A. pichinchensis*. The airlift bioreactor significantly improved the production of the compounds from the cell culture. Compound **1** increased its production by 1.82 times on day 7 compared to cultures in flasks whose maximum production occurred on day 8. Likewise, cultures in the dark showed that the production of compounds in the reactor showed an increase of 2.86 times in 7 days, while this was 8 days for the flask cultures. On the other hand, compound **2** produced 1.36 times the compound of day 14 in the reactor in relation to the production monitored on day 16 of cultures in flasks under the photoperiod. Additionally, compound production in the reactor showed a 1.42-fold improvement over quantification in flask cultures under dark on day 16.

These results highlight that cell suspension cultures of *A. pichinchensis* can be scalable to bioreactors to increase the production of bioactive compounds. In other studies, an increase in the production of compounds in the bioreactor has been reported. For instance, cell suspension cultures of *Scrophularia striata* revealed that phenylethanoid glycosides and phenolic compounds were produced in greater amounts when grown in bioreactors rather than flask cultures [100,101]. Suspension cells of *Prunella vulgaris* showed increased production of antioxidant compounds when cultured under different photoperiod regimes and sucrose concentrations in an airlift bioreactor compared to flask cultures [79].

Likewise, the production of ajmalicin in *Catharanthus roseus* cells cultured in an airlift bioreactor was increased on day 14 [102]. On the other hand, the shikonin production of *Arnebia* sp. was improved in an airlift bioreactor under dark conditions and pH 5.75 [103]. The protein brazzein was produced and its production was even improved by transgenic *Daucus carota* cells that were grown in an airlift bioreactor compared to animal cells that also produce the protein, which opens the panorama of plant cell applications [104]. Cell density and ginseng production of *Panax notoginseng* have also been dramatically increased when cultured in an airlift reactor after 15 days of culture [105]. It has also been reported that the addition of CO_2_ and ethylene to suspension cultures of *Thalictrum rugosum* grown in an airlift bioreactor increased alkaloid production [106].

*Ocimum basilicum* cells grown in a 2.5 L airlift bioreactor increased rosmarinic acid accumulation compared to those grown in a flask [107]. The production of paclitaxel and baccatin III from suspension cell cultures of *Taxus wallichiana* was successfully carried out in a 20 L airlift bioreactor, significantly improved compared to flasks [97]. Squalene has also been produced prominently in *Santalum album* cultures grown in an airlift bioreactor compared to those in flasks [108]. On the other hand, the production of antifungal saponins from *Solanum chrysotrichum* plant cells increased when they were grown in an airlift bioreactor compared to the yields found in flask cultures [98].

## 3. Materials and Methods

### 3.1. Obtaining Plant Material

Cell suspension cultures (CSC) of *Ageratina pichinchensis* were previously established in shake flasks by our working group [62]. To increase biomass, CSCs were subcultured every 15 days in 500 mL Erlenmeyer flasks containing 100 mL of liquid MS [109] basal medium with an inoculum size of 10% (*v*/*v*). Sterile MS contained 30 g/L sucrose, 1.0 mg/L α-naphthaleneacetic acid (NAA), and 0.1 mg/L furfurylaminopurine (KIN) at pH 5.7. Cultures were maintained under the same shaking and incubation conditions [62].

### 3.2. Cell Culture in Shake Flask under Photoperiod and Absolute Darkness Conditions

CCS were cultured under photoperiod and absolute darkness (absence of light during growth kinetics) conditions to compare the production of 2,3-dihydrobenzofuran (**1**) and 3-epilupeol (**2**) compounds. For both incubation conditions, growth kinetics were performed using several 250 mL flasks using the same culture medium composition. Each flask containing 50 mL of MS medium was inoculated with 3 g of cells and incubated on an orbital shaker at 120 rpm at 25 ± 2 °C under photoperiod (16 h light/8 h darkness with 50 µM m^−2^ s^−1^ fluorescent light intensity) or under absolute darkness. Subsequently, three flasks were harvested every two days, and the culture was allowed to grow for 22 days. Biomass was washed with distilled water, filtered through a cellulose filter (Whatman No. 1) using a vacuum pump, and then dried in an oven at 55 °C until reaching a constant weight. Dry weight data were recorded and used to perform the growth curve. The specific growth rate was calculated using the natural logarithm (ln) of biomass dry weights of the exponential phase versus time, the doubling time was determined with the equation dt = ln2/µ. In addition, cell viability was monitored during culture growth and samples of the filtered cell culture medium were also obtained to determine the content of total sugars as mentioned below.

### 3.3. Bioreactor

#### 3.3.1. Bioreactor Characteristics

Airlift bioreactors (ALB) are commonly used for suspension culture of plant cells [110]. In this study, a 2 L capacity airlift bioreactor (Applikon Holland) was used (Figure 7), equipped with a suction tube placed in the center of the reactor through which air was administered through a diffuser using a suction pump submersible water (Zhiquan CMH042, Eiko Co. Ltd., Shanghai, China). The pH was measured using an InPro4260/SG/225/PT100 electrode (Mettler-Toledo, New York, OH, USA). The air flow rate was controlled with a Rotameter (Total Temperature Instrumentation, Inc., Vermont, USA). A MasterFlex^®^ 77601-10 peristaltic pump was used to feed the culture medium and obtain the samples.

#### 3.3.2. Cell Cultures and Bioreactor Operating Conditions

Culture medium consisted of sterile MS supplemented with 30 g/L sucrose, 1.0 mg/L NAA, 0.1 mg/L KIN, and pH value was adjusted to 5.7. The bioreactor was operated with a total volume of 1.7 L. To characterize cell culture growth, the bioreactor was inoculated with fresh cell biomass (5%, *w*/*v*) with 10 days of growth in flask (exponential phase). The bioreactor was maintained in an incubation room at 25 ± 2 °C under photoperiod (16 h light/8 h darkness with 50 µM m^−2^ s^−1^ fluorescent light intensity). The pH values were maintained at 5.5–5.8 by adding NaOH solution at 0.1 M. The bioreactor was operated at 0.5 volumes of air/medium volume/min (vvm) incorporating a sterile gas with an air pump through a flow meter and a sterile filter (0.2 µm).

### 3.4. Cell Viability in Shake Flask and Bioreactor

The cell viability was determined for both cell cultures grown in shake flasks and bioreactor. Viability was determined considering the integrity of the membrane by using Evan’s blue at 0.25% [111]. A 1 mL sample of cell suspension was obtained from each flask or bioreactor, mixed with the dye, incubated for 5 min, and then cell counted. Viable cells were considered those that were not stained. All experiments were repeated three times, with three replicates each.

### 3.5. Sugar Quantification in Shake Flask and Bioreactor

Sugar quantification was determined in the culture media of both cells grown in flasks and bioreactor. The culture medium filtered from the cell suspension cells of the growth kinetics was used to determine the total sugars by the phenol–sulfuric method [112]. For each sampling of growth kinetics an aliquot of 250 µL of culture medium was diluted in distilled water, a 500 µL aliquot was obtained and 500 µL of phenol 5% (*w*/*v*) was added in digester tubes and placed in a rack submerged in a cold water bath. Then, 5 mL of concentrated sulfuric acid was added to the mixture and allowed to stand for 15 min and analyzed in a spectrophotometer at 490 nm against a blank. A calibration curve was performed using sucrose as a standard at concentration of 0.1 to 1.0 µg/mL.

### 3.6. Extraction of 2,3-Dihydrobenzofuran and 3-Epilupeol of Cell Cultures

Both the biomasses of the growth kinetics harvested at days 0, 2, 4, 6, 8, 10, 12, 14, 16, 18, 20, and 22 of the growth kinetics in shake flasks, and at days 0, 1, 2, 3, 4, 5, 6, 7, 8, 9, 10, 11, 12, 13, 14, 15, and 16 of the growth kinetics in airlift reactor were filtered, washed and dried in an oven at 55 °C until constant weight. Each sample was extracted three times by sonication with 25 mL ethyl acetate, then concentrated on a rotary evaporator.

### 3.7. Quantification of 2,3-Dihydrobenzofuran and 3-Epilupeol by GC-MS

Extracts were analyzed in an HP Agilent Technologies 6890 gas chromatograph equipped with an MSD 5973 quadrupole mass detector (HP Agilent, Markham, ON, Canada), equipped with a capillary column HP-5MS (length: 30 m; inside diameter: 0.25 mm; film thickness: 0.25 µM). The helium carrier gas was set to the column (1 mL per minute at constant flow). The inlet temperature was set at 250 °C while oven temperature was initially at 40 °C (held for 1 min) and increased to 280 at 10 °C/min. The mass spectrometer was operated in positive electron impact mode with ionization energy of 70 eV. Detection was performed in selective ion-monitoring (SIM) mode and peaks were identified and quantitated using target ions. Quantification was performed in triplicate using a calibration curve of 2,3-dihydrobenzofuran (**1**) at 2.2, 1.1, 0.55, 0.275, 0.1375, and 0.06875 mg/mL and 3-epilupeol (**2**) at 0.350, 0.175, 0.0875, 0.04375 and 0.02187 mg/mL as reference standards, previously purified and characterized by our work group [62]. Calibration curve showed acceptable linearity with correlation coefficients r^2^ = 0.9926 and 0.9997, respectively. The quantification of compounds in the extracts was expressed in terms of µg per g of biomass dry (µg/g DW).

### 3.8. Statistical Analysis

All dates are presented as the mean values ± standard deviation of at least three replicates. Growth parameters and the production of compounds **1** and **2** were compared by means of one-way of analysis of variance (ANOVA), factors with a *p* ≤ 0.05 were considered significant. Significant differences in the experiments were calculated by Tukey test (GraphPad Prism^®^ version 9.3.1 software, Dotmatics, San Diego, CA, USA).

## 4. Conclusions

*A. pichinchensis* has been studied from different approaches due to its ethnomedical uses, which makes it interesting for the development of crops that allow sustainable production of its bioactive compounds. In this work, a laboratory-scale bioreactor of a cell culture of *A. pichinchensis* is reported for the first time. We found that photoperiod conditions were more suitable than absolute darkness to produce biomass and compounds **1** and **2** in both production systems for *A. pichinchensis*. In addition, the production of compounds in an airlift bioreactor showed a significant increase in the production of both compounds (1.82 and 1.35 folds, respectively) under photoperiod conditions, and the production time was reduced by half. This work opens an encouraging path to improve the production of these compounds using other biotechnological alternatives, such as the use of elicitors using this system, which can contribute to the development of sustainable alternatives for the use of medicinal plants and is currently a priority need due to the rapid reduction in wild populations.

## Figures and Tables

**Figure 1 molecules-28-00578-f001:**
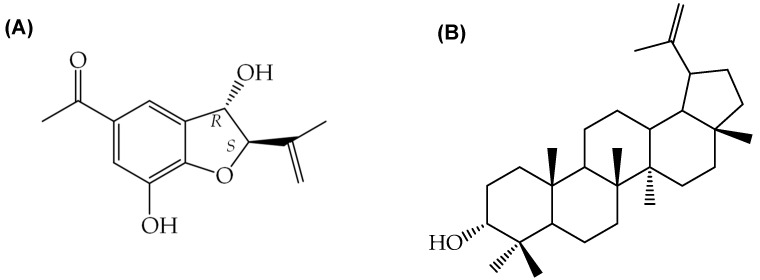
Anti-inflammatory compounds isolated from *A. pichinchensis* cell cultures. (**A**) (*2S*,*3R*)-5-acetyl-7,3α-dihydroxy-2β-(1-isopropenyl)-2,3-dihydrobenzofuran (**1**); (**B**) 3-epilupeol (**2**).

**Figure 2 molecules-28-00578-f002:**
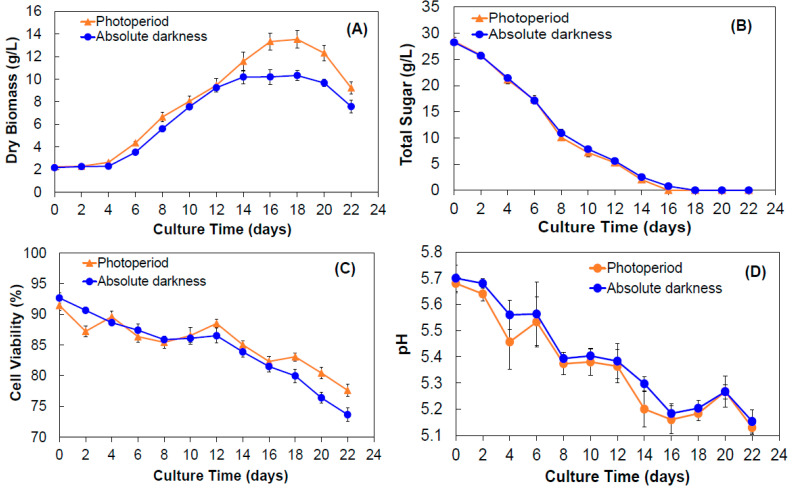
Growth kinetics of *A. pichinchensis* cell culture produced in shaken flasks under photoperiod and absolute darkness conditions for 22 days of culture. (**A**) Yield of dry biomass; (**B**) sucrose consumption; (**C**) cell viability; (**D**) pH values of the culture medium.

**Figure 3 molecules-28-00578-f003:**
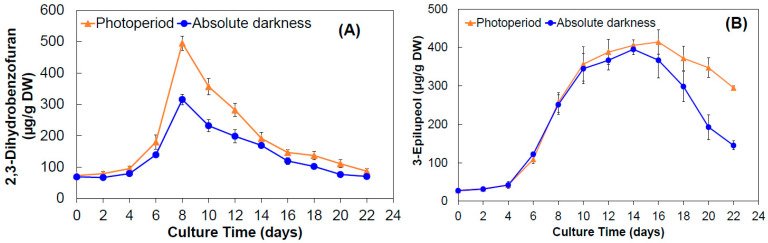
Production kinetics of compounds in cellular biomass of *A. pichinchensis* cultured in shaken flasks under photoperiod or absolute darkness for 22 days. (**A**) 2,3-dihydrobenzofuran (**1**); (**B**) 3-epilupeol (**2**).

**Figure 4 molecules-28-00578-f004:**
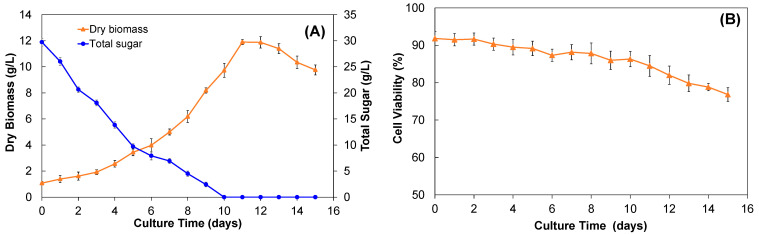
(**A**) Growth kinetics and (**B**) viability of cell culture of *A. pichinchensis* grown in an airlift bioreactor for 15 days under photoperiod conditions.

**Figure 5 molecules-28-00578-f005:**
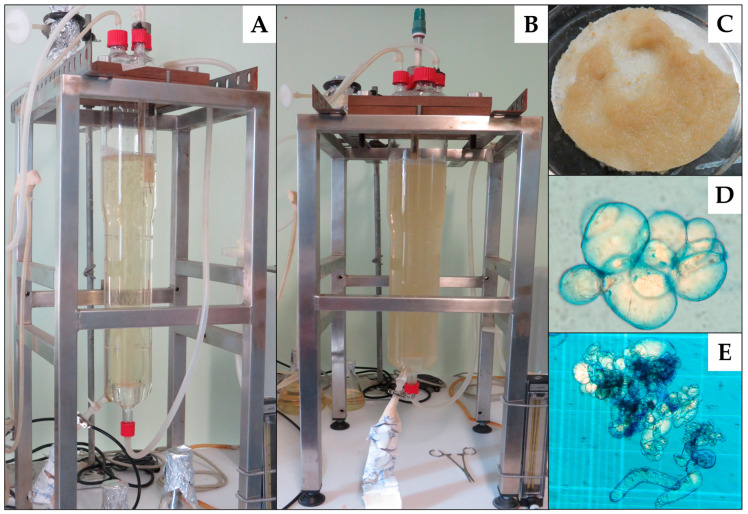
Cell culture of *A. pichinchensis* produced in an airlift bioreactor. (**A**) Cell culture during the first three days of growth; (**B**) cell culture after 4 days; (**C**) biomass appearance of exponential phase; (**D**) micrograph of the cell culture at 6 days (40X); (**E**) micrograph of the cell culture at 12 days (10X).

**Figure 6 molecules-28-00578-f006:**
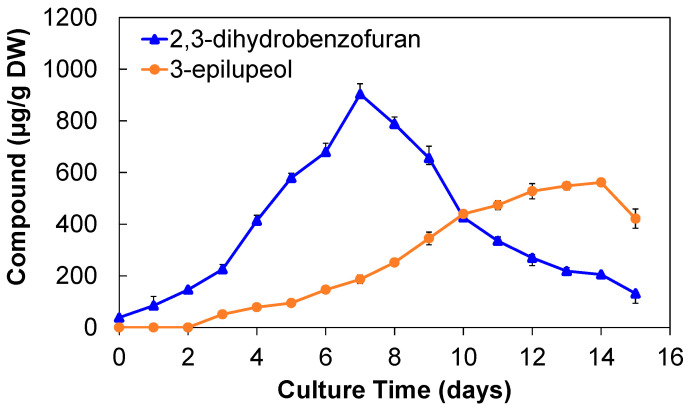
Kinetics of 2,3-dihydrobenzofuran (**1**) and 3-epilupeol (**2**) production in a cell culture of *A. pichinchensis* grown in an airlift bioreactor for 15 days.

**Figure 7 molecules-28-00578-f007:**
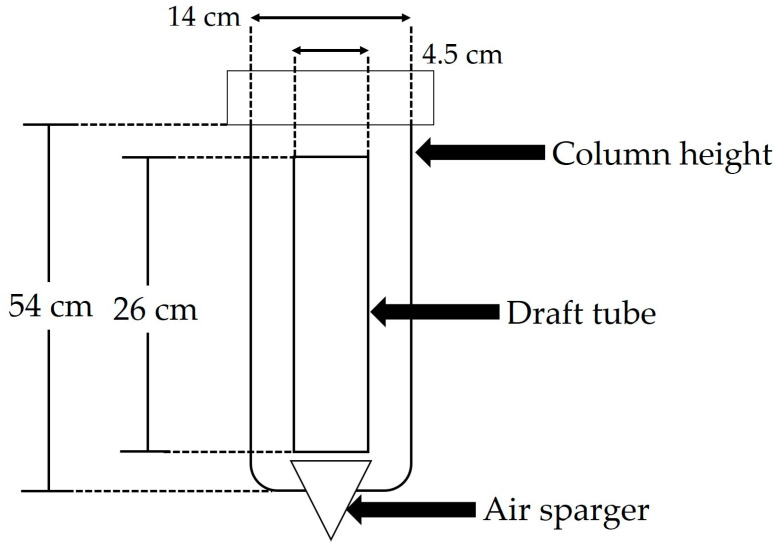
Scheme of an airlift bioreactor used for cell culture of *A. pichinchensis*.

**Table 1 molecules-28-00578-t001:** Comparison of 2,3-dihydrobenzofuran and 3-epilupeol content in different in vitro culture systems of *A. pichinchensis* on specific days of maximum production.

Culture System	Culture Time (Day)	2,3-Dihydrobenzofuran (µg/g DW)	Culture Time (Day)	3-Epilupeol (µg/g DW)
Callus culture in jars/photoperiod *	30	650.00 ± 11.00	30	201.10 ± 15.00
Cell culture in flasks/photoperiod	8	495.04 ± 22.85	16	414.24 ± 31.56
Cell culture in flasks/absolute darkness	8	315.44 ± 16.72	16	395.14 ± 13.32
Cell culture in airlift bioreactor/photoperiod	7	903.02 ± 41.06	14	561.63 ± 10.16

* Reference: [61].

## Data Availability

Not applicable.

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
