# Peer review of "Obtaining 2,3-Dihydrobenzofuran and 3-Epilupeol from Ageratina pichinchensis (Kunth) R.King & Ho.Rob. Cell Cultures Grown in Shake Flasks under Photoperiod and Darkness, and Its Scale-Up to an Airlift Bioreactor for Enhanced Production"

_molecules, 2023, doi:10.3390/molecules28020578_

Round 1

Reviewer 1 Report

I reviewed the manuscript entitled "Obtaining 2,3-Dihydrobenzofuran and 3-Epilupeol from Ageratina pichinchensis Cell Cultures Grown in Shake Flasks under Photoperiod and Darkness, and its Scale-Up to an Airlift Bioreactor for Enhanced Production" in which the authors have evaluated the production of the anti-inflammatory compounds.

The paper is acceptable however it needs some corrections and modifications before publication. 

1. Introduction part, the scientific name of the plant sample should not be short. The first time, they should be complete and with the author's name (Ageratina pichinchensis (Kunth) R.M.King&H.Rob.)

2. What is the importance of 2,3-dihydrobenzofuran and 3-epilupeol? Why did the authors interest to produce these compounds? Add a short background of these compounds to the introduction.

3. In Figure 5, the letter (A, B, C, D, and E) of the explanatory title should correlate with the letter in the figure.

4. Please check the consistency of the word (airlift and air-life). I found two versions of this word, for example, in P5 L189.

5. Why did the authors use GC-MS for the quantification of 2,3-dihydrobenzofuran and 3-epilupeol?

Author Response

Reviewer 1

Point 1. I reviewed the manuscript entitled "Obtaining 2,3-Dihydrobenzofuran and 3-Epilupeol from Ageratina pichinchensis Cell Cultures Grown in Shake Flasks under Photoperiod and Darkness, and its Scale-Up to an Airlift Bioreactor for Enhanced Production" in which the authors have evaluated the production of the anti-inflammatory compounds.

The paper is acceptable however it needs some corrections and modifications before publication.

Introduction part, the scientific name of the plant sample should not be short. The first time, they should be complete and with the author's name (Ageratina pichinchensis (Kunth) R.M.King&H.Rob.)

Response 1. This was done. The full name was modified according to https://www.gbif.org/es/species/5400249. Lines 3, 22 and 63.

Point 2. What is the importance of 2,3-dihydrobenzofuran and 3-epilupeol? Why did the authors interest to produce these compounds? Add a short background of these compounds to the introduction.

Response 2. This was done. Lines 71-78 have been rewritten.

Point 3. In Figure 5, the letter (A, B, C, D, and E) of the explanatory title should correlate with the letter in the figure.

Response 3. The paragraph was rewritten. Lines 175-179 and 189-190.

Point 4. Please check the consistency of the word (airlift and air-life). I found two versions of this word, for example, in P5 L189.

Response 4. This was done. The entire manuscript was reviewed and the word “Air-lift” was changed by “airlift”.

Point 5. Why did the authors use GC-MS for the quantification of 2,3-dihydrobenzofuran and 3-epilupeol?

Response 5. In previous studies, we isolated and purified the compounds 2,3-dihydrobenzofuran and 3-epilupeol, for which we used different spectroscopic techniques, in addition we standardized the methodology to quantify these compounds by GC-MS (https://doi.org/10.3390/molecules23061258; https://doi.org/10.3390/plants9101398. In those studies, our method was reproducible, then, we decided to use the same method and equipment to identify and quantify the compounds in this study.

Reviewer 2 Report

I suggest sending the article to a journal whose profile more closely matches the research conducted.

Some of the inaccuracies that shall be amended:

Abstract

Symbols such as “µ” and “td” shall be explained when appear for the first time, like in the following sentence: “Among kinetic parameters, µ was 0.2216 days-1 and td was 3.13 days.”

All typo mistakes shall be removed from the manuscript.

Results and Discussion section:

The term "Total Darkness" is too colloquial or suggests absolute darkness, it should be changed to be more appropriate.

The units shall be unified. It shouldn’t like that” “ Higher production of 2,3-dihydrobenzofuran was achieved with photoperiod at day 8, reaching a maximum production of 495.04 ± 22.85 µg/g DW (Figure 3A). Despite  the same behavior occurred under dark conditions the maximum yield at day 8 was re-  duced by 36.3%.”

Author Response

Reviewer 2

Point 1. I suggest sending the article to a journal whose profile more closely matches the research conducted.

Response 1. We reviewed the special issue “Discovery of Bioactive Ingredients from Natural Products III”, and our study matches the scope of this journal. Moreover, some related works have been published https://doi.org/10.3390/molecules27238508; https://doi.org/10.3390/molecules27248679.

Point 2. Some of the inaccuracies that shall be amended:

Abstract

Symbols such as “µ” and “td” shall be explained when appear for the first time, like in the following sentence: “Among kinetic parameters, µ was 0.2216 days-1 and td was 3.13 days.”

Response 2. This was done. The full name and its abbreviation were included in parentheses in all the necessary words.

Point 3. All typo mistakes shall be removed from the manuscript.

Results and Discussion section:

The term "Total Darkness" is too colloquial or suggests absolute darkness, it should be changed to be more appropriate.

Response 3. The entire manuscript was revised again. The term “Total darkness” was substituted by “Absolute darkness”.

Point 4. The units shall be unified. It shouldn’t like that” “ Higher production of 2,3-dihydrobenzofuran was achieved with photoperiod at day 8, reaching a maximum production of 495.04 ± 22.85 µg/g DW (Figure 3A). Despite the same behavior occurred under dark conditions the maximum yield at day 8 was reduced by 36.3%.”

Response 4. Change was made to the expressions in lines 121-123.

Round 2

Reviewer 2 Report

The term "absolute darkness" should be defined in the “Cell Culture in Shake Flask Under Photoperiod and Absolute Darkness Conditions” section - otherwise it's not known exactly how "absolute darkness" differs from culturing plant material in the dark, as it is usually done.

Author Response

Point 1. The term "absolute darkness" should be defined in the “Cell Culture in Shake Flask Under Photoperiod and Absolute Darkness Conditions” section - otherwise it's not known exactly how "absolute darkness" differs from culturing plant material in the dark, as it is usually done.

Response 1. The change was made in lines 260-261.